# Expression of Interferons Lambda 3 and 4 Induces Identical Response in Human Liver Cell Lines Depending Exclusively on Canonical Signaling

**DOI:** 10.3390/ijms22052560

**Published:** 2021-03-04

**Authors:** Mariia Lunova, Jan Kubovciak, Barbora Smolková, Mariia Uzhytchak, Kyra Michalova, Alexandr Dejneka, Pavel Strnad, Oleg Lunov, Milan Jirsa

**Affiliations:** 1Institute for Clinical & Experimental Medicine (IKEM), 14021 Prague, Czech Republic; miji@ikem.cz; 2Institute of Molecular Genetics of the Czech Academy of Sciences, 14220 Prague, Czech Republic; kubovcij@img.cas.cz; 3Department of Optical and Biophysical Systems, Institute of Physics of the Czech Academy of Sciences, 18221 Prague, Czech Republic; smolkova@fzu.cz (B.S.); uzhytchak@fzu.cz (M.U.); dejneka@fzu.cz (A.D.); lunov@fzu.cz (O.L.); 4Institute of Medical Biochemistry and Laboratory Diagnostics, General University Hospital and 1st Faculty of Medicine of Charles University, 12808 Prague, Czech Republic; kyra.michalova@lf1.cuni.cz; 5Department of Internal Medicine III, University Hospital RWTH (Rheinisch-Westfälisch Technische Hochschule) Aachen, 52062 Aachen, Germany; pstrnad@ukaachen.de

**Keywords:** interferon stimulated genes, IFNLR1, IL10R2, knockout, transcriptome

## Abstract

Lambda interferons mediate antiviral immunity by inducing interferon-stimulated genes (ISGs) in epithelial tissues. A common variant rs368234815TT/∆G creating functional gene from an IFNL4 pseudogene is associated with the expression of major ISGs in the liver but impaired clearance of hepatitis C. To explain this, we compared Halo-tagged and non-tagged IFNL3 and IFNL4 signaling in liver-derived cell lines. Transfection with non-tagged IFNL3, non-tagged IFNL4 and Halo-tagged IFNL4 led to a similar degree of JAK-STAT activation and ISG induction; however, the response to transfection with Halo-tagged IFNL3 was lower and delayed. Transfection with non-tagged IFNL3 or IFNL4 induced no transcriptome change in the cells lacking either IL10R2 or IFNLR1 receptor subunits. Cytosolic overexpression of signal peptide-lacking IFNL3 or IFNL4 in wild type cells did not interfere with JAK-STAT signaling triggered by interferons in the medium. Finally, expression profile changes induced by transfection with non-tagged IFNL3 and IFNL4 were highly similar. These data do not support the hypothesis about IFNL4-specific non-canonical signaling and point out that functional studies conducted with tagged interferons should be interpreted with caution.

## 1. Introduction

Interferons are secreted in response to viral infections and mediate antiviral immunity by inducing expression of interferon-stimulated genes (ISGs). Expression of ISGs is triggered by interferons binding to the membrane receptors, which are specific for each of the three distinct interferon types (type I, II and III) [1]. Interferon binding to the high affinity receptor subunit triggers association with the low affinity receptor subunit, formation of the ternary or hexameric signaling complex specific for type I and III or type II interferons and subsequent activation of kinases JAK1, JAK2 and TYK2 associated with receptor subunits. Activated kinases mediate phosphorylation of STAT proteins, phosphorylated STATs dimerize and the dimers act either as a transcription factors themselves or associate with IRF-9 thereby forming a ternary transcriptional complex named ISGF-3 [1].

Three type III (lambda) interferons named IFNL1 (IL28A), IFNL2 (IL-29) and IFNL3 (IL28B) are secreted by a broad number of immune and epithelial cells [2,3]. Their signal is transduced through a heterodimeric receptor composed of IFNLR1 and IL10R2 subunits [1,2,3]. Since IFNLR1, the interferon type III high affinity partner that is essential for canonical IFNL signaling, is expressed almost exclusively in epithelial cells, type III interferons play an important role in the antiviral defense of epithelial cells including the liver [4,5].

A cluster of linked non-coding single nucleotide polymorphisms (rs12979860, rs8099917 and several others) located upstream of *IFNL3* has been identified as the genetic factor affecting the clearance of hepatitis C virus (HCV) in four independent genome wide association studies [6,7,8,9]. In a subsequent study [10], Prokunina-Olsson et al. demonstrated that a variant ∆G of a dinucleotide polymorphism rs368234815 (TT or ∆G, formerly ss469415590), which is in high linkage disequilibrium with rs12979860 (C or T), creates an open reading frame of the type III interferon gene named *IFNL4* and is associated with impaired spontaneous and interferon alpha-induced clearance of HCV. Interestingly, untreated HCV patients carrying the *IFNL3* rs12979860 T allele had higher hepatic expression levels of ISGs than GG homozygotes, but no further increase was seen after exposure to exogenous interferon alpha [11,12]. Similar observations were made in IFNL4-transfected HepG2 cells and HCV-infected primary hepatocytes homozygous for *IFNL4* rs368234815 ∆G [10,13].

The mechanism by which IFNL4 impairs HCV clearance remains unclear. Prokunina-Olsson et al. [10] conducted experiments with recombinant IFNL3 and IFNL4 produced in the Sfs9 baculoviral system and observed that stimulation of HepG2 cells with recombinant purified IFNL3, interferon alpha or transfection with full length Halo-tagged *IFNL4* activated the ISRE-Luc and IRF1-Luc reporter. Notably, no activation was seen after transfection with Halo-tagged shorter splice variants of *IFNL4.* Since transient transfection with the construct expressing *IFNL4* decreased HCV RNA replication, the authors suggested that IFNL4 may activate JAK-STAT signaling non-canonically through an IFNL4-specific receptor consisting of IFNLR1 and a yet undefined second receptor subunit.

Hamming et al. [14] suggested that both recombinant proteins, IFNL3 and IFNL4, produced in *E. coli* stimulate ISG expression through the same canonical IFNLR1/IL10R2 receptor, since markedly diminished ISG responsiveness was found in HEK293 cells expressing low levels of IFNLR1 and in IFNLR1-transfected HEK293 cells after IL10R2 blockage via an anti-IL10R2 antibody or *IL10R2* siRNA silencing. Both recombinant interferons exerted the same antiviral activity in HuH7-Lunet hCD81-Fluc cells transfected with the HCV genome. Interestingly, Hamming et al. [14] also noted that, in contrast to MYC- and FLAG-tagged IFNL3, MYC- and FLAG-tagged IFNL4 is retained in the Golgi apparatus due to its impaired glycosylation. This prompted investigators to speculate about a non-canonical pathway using a distinct IFNL4-specific intracellular receptor [15,16]. However, owing to the fact that lower ISG responsiveness to IFNL3 and IFNL4 was observed after blocking/downregulation of IL10R2 [14,17], IFNLR1-deficient HepG2 cells did not respond at all [18], and the transcriptome analysis of IFNL3 and IFNL4 protein-stimulated human primary hepatocytes did not reveal genes specifically and significantly regulated by IFNL4 [19]; the existence of such an alternative pathway was challenged.

Since even the most current in vitro data suggest that IFNL4 has antiviral properties [20], while it impairs virus clearance and displays “anti-inflammatory” function in vivo [15,21,22], we attempted to find the anticipated non-canonical IFNL4-specific intracrine signaling circumventing the JAK-STAT pathway in cells lacking either IFNLR1 or IL10R2. Moreover, we checked whether canonical JAK-STAT signaling could be modified by IFNL4 expression in cytosol.

## 2. Results

### 2.1. Transient Transfection with Tagged and Non-tagged IFNL3 and IFNL4 Induced Similar mRNA Levels of Interferons but Different Levels of Interferon-Stimulated Genes

Consistent with the previously published data [10,23], IFNL3 and IFNL4 were not detectable in 48 h mock-transfected HuH7 cells by RT-PCR and confocal microscopy (Figure 1A,B and Appendix A). The cells transiently transfected with Halo-tagged and non-tagged forms of IFNL3 or IFNL4 showed similar mRNA expression levels of the corresponding interferons (Figure 1A).

To compare the activity of tagged and non-tagged IFNL3 and IFNL4, we assessed expression levels of major ISGs in 48h transfected cells. Non-tagged IFNL3 and IFNL4 activated expression of *MX1*, *ISG15*, *IFI6*, *USP18, RSAD2, IFI27, IFITM1* and *OAS1* to the same extent in HuH7 cells (Figure 1C, Appendix A). Cells transfected with tagged IFNL4 reached similar expression levels of *MX1* and *IFI6,* but levels of *ISG15* and *USP18* were somewhat lower. Contrarily, cells transfected with tagged IFNL3 had significantly lower expression levels of all four ISGs (Figure 1C).

To find out the time course of non-tagged *IFNL3* and *IFNL4* mRNA expression, we determined their expression levels in HuH7 cells at 8, 24, 32, 48 and 56 h from transfection (Figure 2A). The data showed the same levels of *IFNL3* and *IFNL4* mRNA at 8, 32 and 48 h; at 24 h we observed somewhat higher IFNL4 expression in IFNL4-transfected cells when compared to IFNL3-transfected cells. Both interferons reached their highest expression levels at 32 h after transfection.

Both non-tagged IFNL3 and IFNL4 induced similar expression levels of *MX1* at 8, 24, 32 and 48 h (Figure 2B). The expression levels of *ISG15*, *IFI6* and *IFI27* did not differ at 32, 48 and 56 h (Figure 2B), all *ISG15*, *IFI6* and *IFI27* showed only slightly higher expression levels in *IFNL4* transfected cells at 24 h after the transfection (Figure 2B).

In the next step, we compared the ability of cells transfected with tagged and non-tagged *IFNL3* and *IFNL4* to activate JAK-STAT signaling by immunoblotting with densitometry quantification (Figure 2C and Appendix A). HuH7 and HepG2 cells were transfected with tagged and non-tagged *IFNL3* or *IFNL4* for 48 h or stimulated with IFNL3 or IFNL4 proteins for 24 h. Transfection time of 48 h was chosen based on our data (Appendix A) and on published data by Prokunina-Olsson et al. Transfection with non-tagged IFNL3 and non-tagged IFNL4 induced similar STAT1 protein expression and phosphorylation (pSTAT1); however, the response to transfection with Halo-tagged IFNL3 was significantly lower when compared to cells transfected with Halo-tagged IFNL4. 

In a separate set of experiments conducted with cells stimulated by non-tagged recombinant interferons, we observed that IFNL3-stimulated cells displayed similar levels of STAT1 and pSTAT1 to those detected in IFNL4-stimulated cells. 

### 2.2. Cytosolic Expression of IFNL3 or IFNL4 Did Not Modify JAK-STAT Signaling

Next, we examined whether IFNL4 retained within the cell is able to interfere with canonical signaling triggered by IFNL3/4. To study that, we transfected HuH7 cells with plasmids carrying incomplete ORFs of non-tagged *IFNL3* or *IFNL4* lacking the code for signal peptides (IFNL3-SP and IFNL4-SP). Neither cytosolic expression of IFNL3 nor cytosolic expression of IFNL4 had any impact on *IFI6*, *MX1*, *ISG15* and *USP18* mRNA expression (Figure 3A). Cytosolic interferons also did not activate STAT1 expression or phosphorylation (Figure 3B). On the other hand, stimulation of cells with cell lysates from IFNL3-SP and IFNL4-SP overexpressing cells showed that both interferons retained their biological activity (Figure 3C).

### 2.3. IFNL3 and IFNL4 Induced Identical Transcriptome Response in Wild Type Cells

To compare the changes induced by IFNL3 and IFNL4 transfection, RNAseq of HuH7 cells transfected for 48 h with non-tagged *IFNL3*, non-tagged *IFNL4* or empty vector was performed.

Both *IFNL3*- and *IFNL4*-transfected cells displayed similar expression profiles as documented by the heat map representing 1000 most expressed genes depicted in Figure 4A. Moreover, we performed generalized linear regression with counts modeled using negative binomial distribution displayed as MA plots (Figure 4B) and found a number of differentially expressed genes between IFNL3 vs. mock and IFNL4 vs. mock transfected cells represented by red dots; however, we did not find a single pathway differentiating IFNL4- from IFNL3-overexpressing cells by KEGG and GO analysis. Only three genes, namely, *IFNL4*, *IFNL3* and *IFNL2,* were found as differently expressed, which is not surprising since transfections induced high expression of *IFNL3* or *IFNL4*. Since *IFNL3* and *IFNL2* are paralogue genes with 98% of cDNA sequence similarities, RNAseq analysis likely assigned some reads originating from *IFNL3* mRNA as *IFNL2*.

### 2.4. IFNLR1 and IL10R2 Deficiency Abolished JAK-STAT Signaling by Both IFNL3 and IFNL4

To explore whether IFNL4 may act independently on the canonical JAK-STAT pathway through interaction with an alternative receptor using none or just one of the two canonical receptor subunits, we generated IFNLR1 and IL10R2-deficient HuH7 and HepG2 cell clones (see Appendix A, Figure 5 and Appendix A for detailed characteristics). Using these clones, we investigated whether IFNL3 or IFNL4 can activate JAK-STAT signaling in the knockout cells. For this purpose, we performed either transfections or protein stimulations of IFNLR1 and IL10R2-deficient cell clones with recombinant interferons (for HuH7 clones see Figure 6, for HepG2 Appendix A). STAT1 and pSTAT1 were detected by immunoblotting in whole cell lysates from cells harvested 48 h after transfection or after 24 h exposure to recombinant interferons. Expression of major ISGs was assessed by RT-PCR. Neither IFNL3 nor IFNL4 activated STAT1 protein expression and phosphorylation in IFNLR1 or IL10R2 knockout cells (Figure 6A, Appendix A). Similarly, neither of the two interferons activated expression of *IFI6*, *MX1* and *ISG15* mRNA (Figure 6B).

To find out whether IFNL4 is able to act through activation of other yet unknown signaling pathway(s) from inside the cell, we performed RNAseq analysis of IFNLR1 and IL10R2-deficient cells transfected with *IFNL4*. Despite several differentially regulated biologically irrelevant genes (for heatmaps see Figure 7), the subsequent data analysis of differentially expressed genes (Appendix A), followed by KEGG and GO analysis, did not detect any signature.

## 3. Discussion

It has become widely accepted that different liver pathologies are associated with inflammation [24,25]. Despite their antiviral activity, IFNLs lack the strong pro-inflammatory effects of type I IFNs and are rather anti-inflammatory and tissue protective. In contrast to the lung, human hepatocytes preferentially generate and respond to IFNLs in a manner similar to respiratory epithelia, whereas dendritic cells, monocyte-derived cells and other cell types play a less important role [26]. Therefore, the detailed characterization of IFNL4 signaling in liver-derived cell lines is important for understanding the antiviral defense mechanisms in the liver. Several studies showed that IFNL4 has higher biological activity than IFNL3 in cells overexpressing the tagged interferons, while cells stimulated with recombinant interferons in the media revealed the same pattern of JAK/STAT activation and ISG expression [23,27]. These and several other functional studies comparing IFNL4 with other lambda interferons were conducted exclusively with large protein-tagged interferons. The fact that the size of large tags such as Halo (33 kDa), GFP (27 kDa) or hemagglutinin (63 kDa) exceeds the size of IFNL3 (22 kDa) or IFNL4 (20 kDa) opens the question about the potential impact of such tags on interferon binding to the receptor and signal transductiongth. Multiple publications have reported that protein tags can alter the biological activity of the studied proteins [28,29], induce changes in protein conformation [30] and lead to undesired effects in structural studies and protein toxicity [31,32]. It has also been shown that substantial amount of Halo-tag can be proteolytically cleaved by cells [33]. Because of that, we compared the activity of tagged and non-tagged interferons and found that tagged IFNL3 has diminished activity when compared to non-tagged IFNL3 (Figure 1C and Figure 2C). Thus our finding may explain the observed discrepancy between identical cell response to protein stimulation by recombinant IFNL3 and IFNL4 but different cell responses to overexpression of tagged IFNL3 and IFNL4 as reported in [23]. However, we could not assess the interferon protein levels in the media due to the difficulties described in that study. 

Our study demonstrated that transfection with non-tagged interferons IFNL3 and IFNL4 equally upregulated STAT1 expression and phosphorylation (Figure 2C, Appendix A) whereas other previously published studies [20,22] have shown that STAT1 expression and phosphorylation are higher in cells overexpressing tagged IFNL4. Furthermore, we analyzed a set of downstream targets of JAK-STAT pathway, namely, *MX1, ISG15, IFI6, USP18, RSAD2, IFI27, IFITM1, OAS1,* and found comparable expression levels induced by both non-tagged IFNL3 and IFNL4 (Figure 1C, Appendix AB). Our data thus imply that IFNL3 and IFNL4 are equally strong inducers of ISGs not only in interferon protein-stimulated but also in transiently transfected cells.

Several studies demonstrated that IFNL4 signaling can be partially blocked by either anti-IFNL4 or anti-IL10R2 antibodies [23,27]. Furthermore, it was shown that both antibody blocking and siRNA silencing of IL10R2 dramatically decreased ISG responsiveness to recombinant proteins IFNL3 and IFNL4 produced in E. coli [14]. However, blocking or siRNA silencing of the receptor does not disprove existence of an alternative receptor or receptor subunit. Therefore, Hong et al. [18] developed an IFNLR1-deficient HuH7 cell line using the CRISPR/Cas9 system and demonstrated that ISRE luciferase reporter activity was completely abrogated by IFNLR1 deficiency. Accordingly, we confirmed that both receptor subunits IFNLR1 and IL10R2 are essential for JAK-STAT signaling by interferons lambda (Figure 7, Appendix A).

In addition, we also showed that cytosolic expression of IFNL3 or IFNL4 lacking signal peptides maintained their biological activity (Figure 3C). However, cytosolic interferons neither activated JAK-STAT signaling from inside the cells, nor modulated JAK/STAT signal triggered by interferons from outside.

RNAseq is a state-of-the-art technique to quantitatively analyze the whole transcriptome [34]. Lauber et al. [19] used this technique to compare transcriptomes of primary human hepatocytes stimulated by recombinant IFNL3 or IFNL4 proteins. The analysis did not reveal any genes that were both specifically and significantly regulated by IFNL4; however, when discussing their findings, the authors admitted that the experimental design could not rule out any additional signaling abilities. For these reasons, we compared transcriptomes of wild type, IFNLR1 and IL10R2 lacking HuH7 cells. Here we show the transcriptomes of wild type HuH7 cells transfected with non-tagged IFNL3 or IFNL4 (Figure 4) and IFNLR1 as well as IL10R2 knockout HuH7 cells transfected by non-tagged IFNL4 or mock (Figure 7). This approach allowed us to detect not only the transduction of the auto- or paracrine signal triggered by secreted interferons from outside of the cells transduced via non-canonical membrane receptor(s), but also any intracrine signal from inside the cells. Our finding of no specific gene signature associated with IFNL4 overexpression, together with the findings by Lauber and colleagues [19], strongly suggests that prevention of HCV clearance by IFNL4 through an alternative signaling pathway is highly improbable.

In conclusion, the observations that neither protein stimulation nor transfection with IFNL4 impacts the transcriptome of cells lacking either of the receptor subunits make the hypothesis about IFNL4-specific non-canonical intra-, auto- or paracrine signaling very unlikely. While we used two well established hepatocellular cell lines, some of the discrepancies might also be due to differing cell culture systems or variations in experimental conditions. The differences between the Halo-tagged and non-tagged IFNL3 activity point out that functional studies conducted with tagged interferons should be interpreted with caution. 

## 4. Materials and Methods

### 4.1. Chemicals and Antibodies

The list of chemicals and antibodies including concentrations used for immunoblotting and immunofluorescence is presented in Appendix A.

### 4.2. Cell Culture

Human liver-derived cell lines Huh7 and HepG2, obtained from the Japanese Collection of Research Bioresources (JCRB, Osaka, Japan) and American Type Culture Collection (ATCC, Manassas, VA, USA), respectively, were cultured in EMEM medium (ATCC, cat. no. 30-2003) supplemented with 10% fetal bovine serum (FBS, cat. no. 10500-064, Thermo Fisher Scientific, Waltham, MA, USA), penicillin, and streptomycin (Sigma, cat. no. P4333, St. Louis, MO, USA) as recommended by the supplier. Cultures were maintained in a humidified 5% CO_2_ atmosphere at 37 °C and the medium was changed twice a week. All cell lines were regularly checked for mycoplasma contamination using the MycoAlert Mycoplasma Detection Kit (Lonza, cat. LT07-418, Basel, Switzerland).

### 4.3. Generation of Knockout Cell Lines

IFNLR1 and IL10R2 knockout cell lines were generated using CRISPR/Cas9 technology. CRISPR/guides selection resource provided by Zhang laboratory (https://zlab.bio/guide-design-resources, accessed on 24 January 2017) was used to design specific guide RNA (gRNA, IFNLR1: 5´-ACTTCAGCGTGTACCTGACA-3′, IL10R2: 5′-CCTCCCGAAAATGTCAGAAT-3′ or 5′-GTGGAGCCTTGGGAGCTGGC-3′) at the 5 prime end of the gene coding regions. Next, gRNA was cloned into the GeneArt® CRISPR Nuclease Vector with OFP Reporter kit (Invitrogen, cat. no. A21174, Carlsbad, CA, USA). HuH7 or HepG2 cells were seeded onto 12-well plates at a density of 400,000 cells per well and transfected by 1µg of CRISPR/Cas9 plasmid the next day. Cell clones expressing OFP, Cas9 and CRISPR were sorted onto a 96-well plate using a BioRad FACS S3e cell sorter 48 h after transfection. The sorted clones were expanded by standard cultivation protocol and the degree of gene editing was assessed at the cDNA, gDNA and protein levels by Sanger sequencing, immunoblotting and immunofluorescence. Clones with complete IFNLR1 or IL10R2 deficiency were selected for further experiments (Appendix A).

### 4.4. Expression Constructs

The expression constructs encoding open reading frames of *IFNL4* (NM_001276254.2), *IFNL3* (NM_172139.2) and those lacking signal peptides (IFNL4-SP, IFNL3-SP) were cloned into the pFC14AHaloTag@CMVFlexi@ mammalian expression vector (Promega, cat. no. G965A, Madison, WI, USA) as described in [10]. All expression constructs were prepared with or without stop codon following the interferon gene ORF that regulates expression of HaloTag. Empty vector was used as a mock control. All plasmids were validated by Sanger sequencing before transfection.

### 4.5. Transient Transfection

For transfection experiments, cells were seeded onto 12-well plates at a density of 400,000 cells per well. The following day, the cells were transfected with 1µg per well of plasmid using Lipofectamine 3000 transfection reagent kit (Thermo Fisher Scientific, cat. no. 23000-008). Briefly, the plasmid was pre-mixed with OPTI-MEM medium (Thermo Fisher Scientific, cat. 31985-062), Lipofectamine reagent 3000 and Enhancer reagent as suggested by manufacturer. The DNA–lipid complex was used to transfect the cells. The transfected cells were harvested 8, 24, 32, 48 and 56 h after transfection for mRNA analysis and 12, 24, 36, 48 and 56 h after transfection for protein extraction.

### 4.6. Interferon Treatment

Cells were seeded onto 12-well plates at a density of 400,000 cells per well. The following day, the cells were treated by recombinant (100 ng/mL) IFNL3 (LifeSpan BioScience, cat. no. LS-G4853, Seattle, WA, USA) or IFNL4 (R&D Systems, cat. no. 9165-IF, Minneapolis, MN, USA). All treatments were performed in four biological replicates. Twenty-four hours after treatment, the cells were collected for analysis.

To determine the activity of IFNL-SP, cells were seeded onto 12-well plates at a density of 400,000 cells per well. The following day, the cells were transfected with IFNL3-SP or IFNL4-SP. After another 48h, the cells were mechanically destroyed and filtered in order to remove cellular debris. The cell lysates were used for immunoblot analysis and cell stimulations. All stimulations were performed in four biological replicates for 24 h.

### 4.7. Real time PCR Analysis

Total RNA was isolated using the Qiagen RNeasy mini kit (cat. no. 74106) followed by DNA removal with the Qiagen Rneasy Kit with on-column Dnase I digestion (cat. no. 79254). The quality and quantity of isolated RNA was assessed using the NanoDrop 8000 instrument (Thermo Fisher Scientific). A total of 2µg of RNA was transcribed into cDNA with a RevertAid H Minus First Strand cDNA Synthesis kit (Thermo Fisher Scientific, cat. no. K1632). Quantitative RT-PCR was performed on an Applied Biosystems Viia7 Real Time PCR system using the Fast Advanced TaqMan Gene expression Master mix (Thermo Fisher Scientific, cat. 4444557) and specific TaqMan Gene Expression Assays listed in Appendix A. Total RNA input for TaqMan assays was 20 ng per reaction. At least 4 biological replicates were analyzed for each group. *GAPDH* expression assay was used as the internal control. The data were analyzed using MS Excel and MaxStat Pro 3.6. Expression was normalized to *GAPDH* expression using the ∆Ct method.

### 4.8. RNA Sequencing and Transcriptome Analysis

For both RNAseq experiments, Poly(A) RNA Selection Kit (Lexogen) was used for mRNA extraction followed by library preparation with SENSE Total RNA-Seq Library Prep Kit (Lexogen, Vienna, Austria). Library size distribution was evaluated on the Agilent 2100 Bioanalyzer using the High Sensitivity DNA Kit (Agilent). Libraries were sequenced on the Illumina NextSeq® 500 instrument using 84bp single-end configuration for the IFNLR1/IL10R2 knockout with transfection analysis and 76bp for the IFNL3/4 induction by transfection analysis. Read quality was assessed by FastQC (http://www.bioinformatics.babraham.ac.uk/projects/fastqc, accessed on 2 September 2019). For subsequent read processing, a bioinformatic pipeline nf-core/rnaseq version (1.3 Phil Ewels. nf-core/rnaseq: nf-core/rnaseq version 1.3. (2019), doi:10.5281/zenodo.2610144), was used separately for each of the analyses. Individual steps included removing sequencing adaptors with Trim Galore (http://www.bioinformatics.babraham.ac.uk/projects/trim_galore/, accessed on 2 September 2019), mapping to reference genome GRCh38 [35] (Ensembl annotation version 95) with HISAT2 [36] and quantifying expression on gene level with feature Counts [37]. Per gene mapped counts served as input for differential expression analysis using DESeq2 R Bioconductor package [26]. Prior to the analysis, genes not expressed in at least two samples were discarded. Shrunken log2-fold changes using the adaptive shrinkage estimator [27] were used for differential expression analysis. The analysis focused on transfection type (IFNL4/3 transfected vs. mock analysis) or knockout with transfection type (IFNLR1/IL10R2 KO, IFNL4 transfected vs. mock analysis) as the variables. Genes exhibiting minimal absolute log2-fold change value of 1 and statistical significance (adjusted *p*-value < 0.05) between compared groups of samples were considered as differentially expressed for subsequent interpretation and visualization. Gene set enrichment analysis was performed using a gene length bias aware algorithm implemented in the goseq R Bioconductor package [28] with KEGG [29] pathways and GO terms [30] data.

### 4.9. Immunoblotting

Total cell lysates were prepared as previously described [38]. Briefly, cells were mechanically homogenized in RIPA buffer (Millipore, cat. no. 20-188, Burlington, NJ, USA) supplemented with a protease inhibitor mix (Promega, cat. no. G6521, Madison, WI, USA) and PhosphoStop (Roche, Basel, Swidzerland), followed by centrifugation to remove non-soluble debris. Protein concentrations were determined using the Pierce BCA protein assay kit (Thermo Fisher Scientific, cat. no. 23225). Equal amounts of protein were separated by 10% SDS–polyacrylamide gel electrophoresis (PAGE) and transferred to PVDF membranes (Invitrogen, cat. LC2005). Membranes were blocked and incubated with the appropriate primary and secondary antibodies (Appendix A). The resulting HRP signal was visualized with the Clarity Max ECL Western Blotting Substrate (Bio-Rad, cat. no. 1705062, Hercules, CA, USA) and detected on the G:Box system (Syngene, Synoptics group, Cambridge, UK).

### 4.10. Immunofluorescence Studies

Control, IFNLR1 and IL10R2 knockout cells were seeded onto 6-channels Ibidi μ-slides (cat. 80606) at a density of 15,000 cells per channel and transfected the next day with 20ng per channel of *IFNL4* or *IFNL3* expressing plasmid. Transfection with empty vector was used as a negative control. After 48 h, cells were stained with HaloTag ligand (Promega, Madison, WI, USA) or the appropriate antibodies. Briefly, cells were washed with PBS, fixed with 4% formaldehyde (Sigma), washed again in PBS, permeabilized with 0.5% Triton X-100 (Sigma) and blocked with 2% BSA (Sigma). After blocking, the cells were incubated with primary antibodies against IFNL4 and IFNL3, washed and stained with Alexa Fluor-conjugated secondary antibodies (Appendix A). Staining for HaloTag was performed in accordance with the manufacturers’ guidelines. Labeled cells were imaged by a high-resolution spinning disk confocal microscope IXplore SpinSR (Olympus, Tokyo, Japan) as described in [38,39]. Fluorescence images were taken with the acquisition software cellSens (Olympus). 

### 4.11. Quantification of IFNLs

ImageJ software (NIH, Bethesda, MD, USA) was used for image processing and fluorescent micrograph quantification. Cellular fluorescence intensity was calculated by normalizing the corrected total cell fluorescence (CTCF) of the full area of interest to average single cell fluorescence. The net average CTCF intensity of a pixel in the region of interest was calculated for each image utilizing a previously described method [40]. 

### 4.12. Statistical Analysis

The results were presented as means ± standard error of mean (SEM). The statistical significance of differences between the groups was determined using ANOVA with subsequent application of the Newman–Keuls test. All statistical analyses were performed using MaxStat Pro 3.6. Differences were considered statistically significant at ***p* < 0.01 and * *p* < 0.05. The statistical analysis of experiments with technical replicates is detailed in figures’ legends.

## Figures and Tables

**Figure 1 ijms-22-02560-f001:**
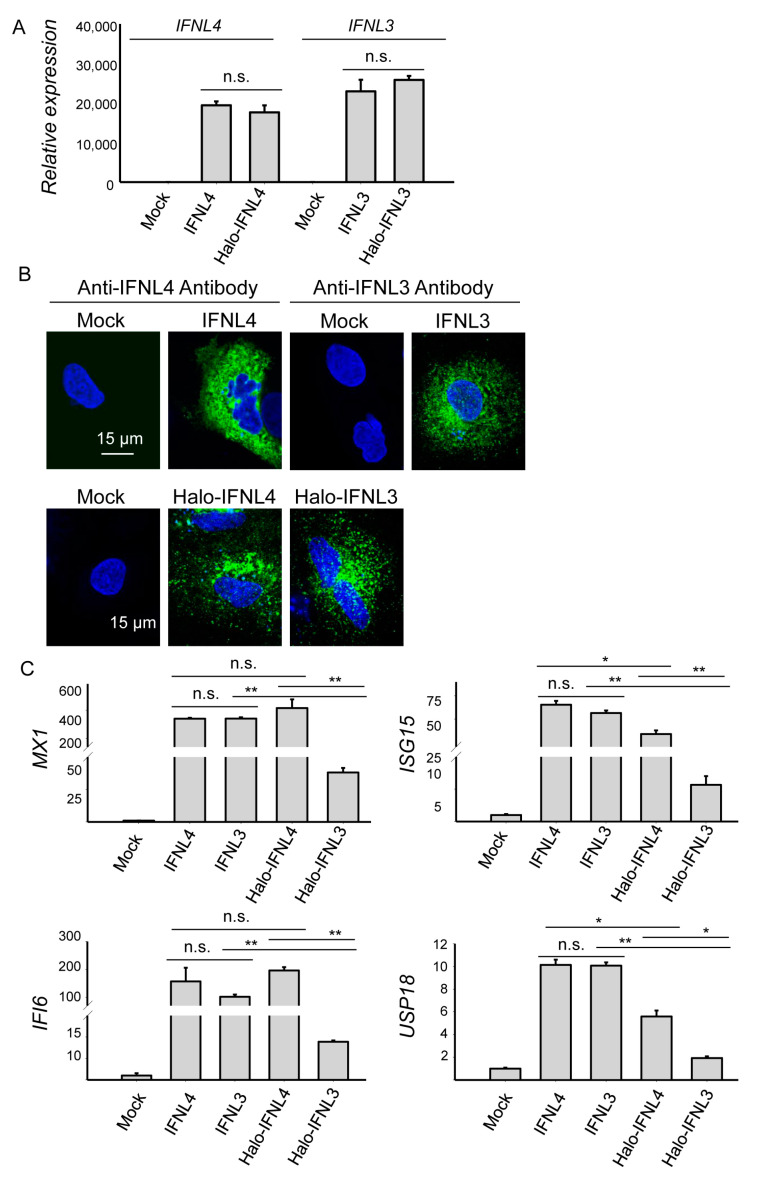
Expression levels of IFNL3, IFNL4 and ISGs in transfected HuH7 cells. (**A**) Relative expression of tagged and non-tagged *IFNL3* or *IFNL4* mRNA in transiently transfected HuH7 cells 48 h after transfection. GAPDH was used as an internal control. Results are presented as mean ± SEM (*n* = 4). (**B**) High resolution confocal images of HuH7 cells transfected with non-tagged (upper series) and tagged (lower series) *IFNL3* or *IFNL4* for 48 h. Empty plasmid was used as a mock control. Blue—nuclei; green—IFNL3 or IFNL4 proteins detected either with the corresponding anti-IFNL3/anti-IFNL4 antibody (non-tagged interferons in upper series) or with anti-Halo tag ligand (tagged interferons in lower series). Scale bar—15 µm. (**C**) Relative expression of *MX1*, *ISG15*, *IFI6*, and *USP18* determined in HuH7 cells 48 h transfected with tagged or non-tagged *IFNL3* or *IFNL4*. *GAPDH* was used as internal control. Results are presented as mean ± SEM (*n* = 4). Differences were considered significant at ** *p* < 0.01 and * *p* < 0.05.

**Figure 2 ijms-22-02560-f002:**
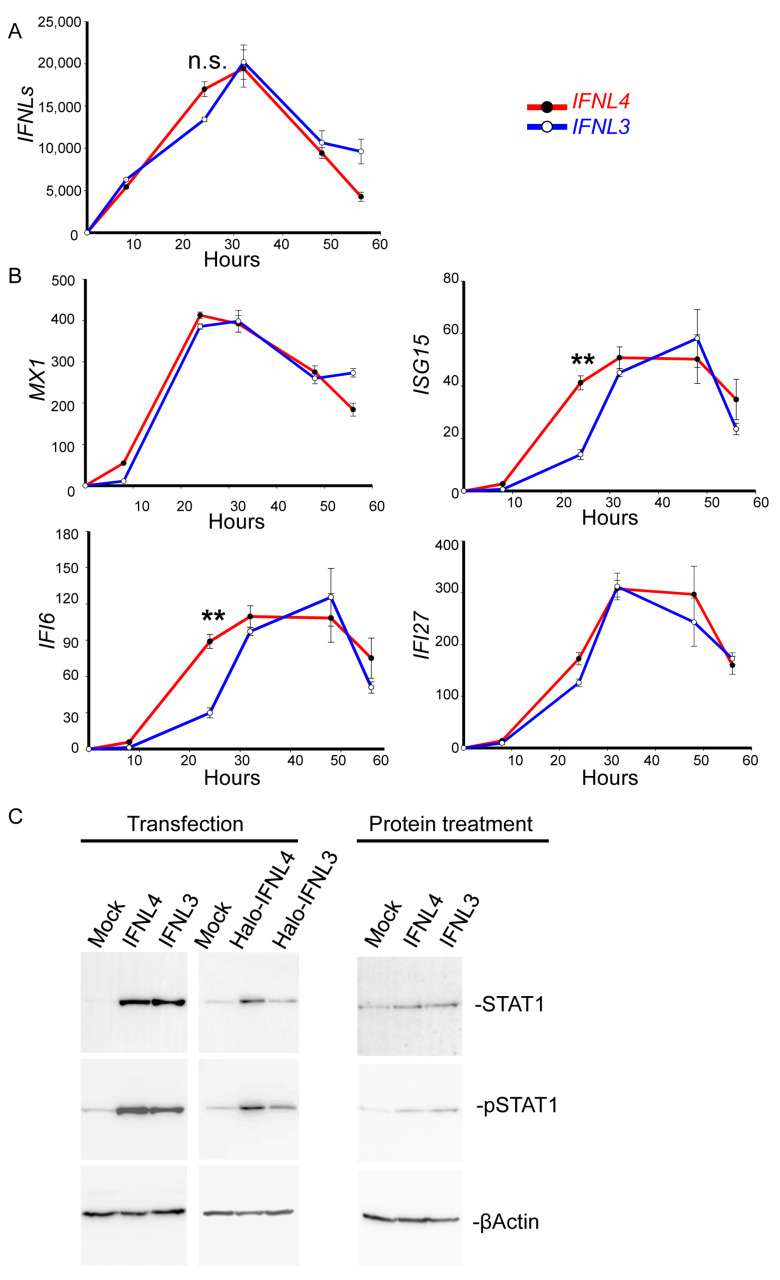
IFNL3 and IFNL4 induce comparable JAK-STAT activation. (**A**,**B**) Kinetics of ISG expression in wild type HuH7 cells transfected with non-tagged *IFNL3* or *IFNL4*. Relative expression of *IFNL3, IFNL4, MX1*, *ISG15*, *IFI6* and *IFI27* calculated to time point 0 is depicted. The data were collected at 8, 24, 32, 48 and 56 h after transfection. The results are presented as a mean ± SEM (*n* = 4). The changes following transfection with IFNL4 and IFNL3 are shown in red and blue, respectively. *GAPDH* was used as internal control. Differences were considered significant at ** *p* < 0.01. Relative expression of ISGs culminated between 32 and 48 h after the transfection. (**C**) The activation of JAK-STAT signaling was determined by immunoblot analysis of wild type HuH7 cells that were either transfected with non-tagged or Halo-tagged *IFNL3* or *IFNL4* for 48 h or stimulated with IFNL3 or IFNL4 proteins for 24 h. Empty plasmid was used as a mock control, ßActin served as the loading control. Equal volumes of 4 biological replicates were loaded in each lane.

**Figure 3 ijms-22-02560-f003:**
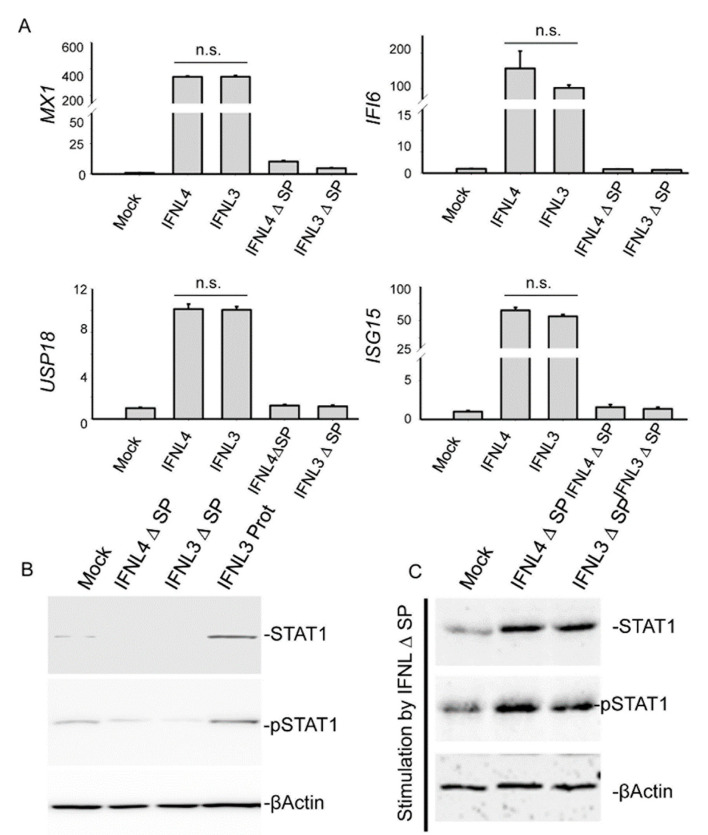
IFNL3 and IFNL4 do not alter JAK-STAT signaling from cytosol. (**A**) Relative expression of *IFI6*, *MX1*, *ISG15* and *USP18* was assessed by real time PCR in HuH7 cells transfected with non-tagged *IFNL3*, *IFNL4*, *IFNL3-SP*, and *IFN4-SP* (SP—signal peptide) for 48 h. GAPDH was used as an internal control. Transfection with empty vector was used as a mock control. The data are presented as mean ± SEM (n ≥ 4). (**B**) Immunoblot detection of STAT1 and pSTAT1 in HuH7 cells transfected with *IFNL3-SP* or *IFNL4-SP*. Transfection with empty vector was used as a mock control. Stimulation with IFNL3 protein for 24 h was used as a positive control (IFNL3 Prot). ßActin represents the loading control. Equal amounts of 4 biological replicates were loaded in each lane. (**C**) Immunoblot detection of STAT1 and pSTAT1 in HuH7 cells which were stimulated for 24 h with cell lysate obtained from cells transfected with *IFNL3-SP* or *IFNL4-SP* for 48 h. ßActin represents the loading control. Equal amounts of 4 biological replicates were loaded in each lane.

**Figure 4 ijms-22-02560-f004:**
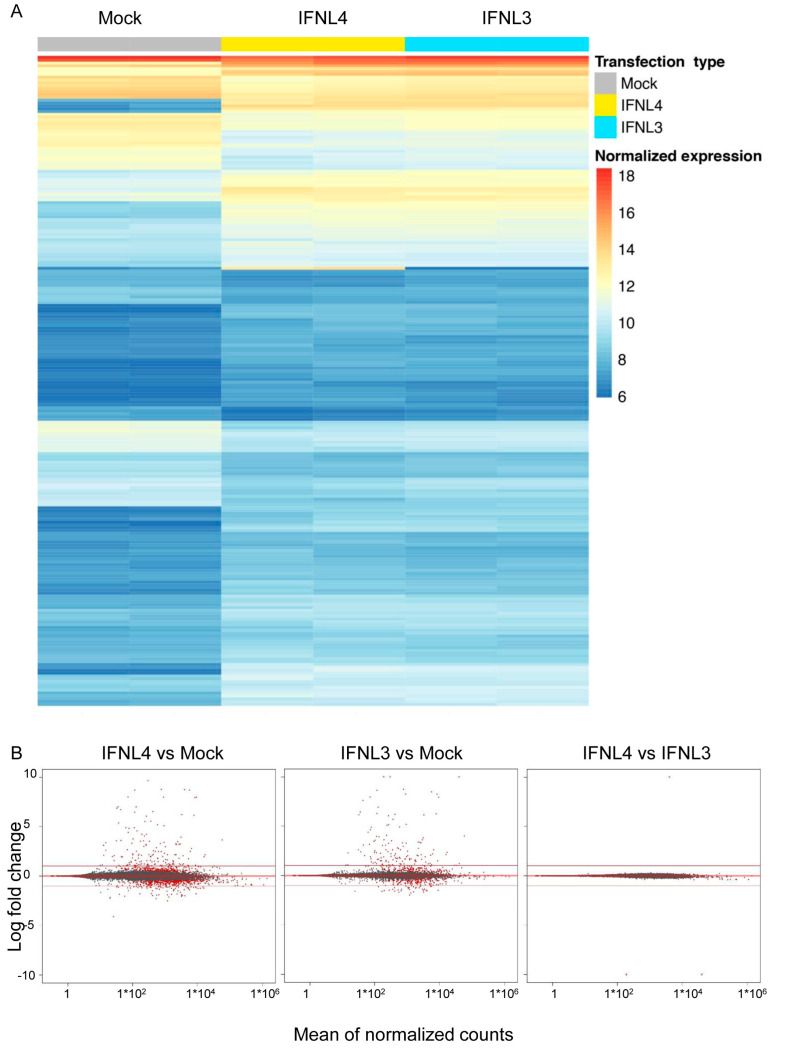
Gene expression profiles of cells expressing *IFNL3* or *IFNL4.* (**A**) Heat map of 1000 most expressed genes determined by RNAseq in HuH7 cells transfected with non-tagged *IFNL3* or *IFNL4* for 48 h. Empty plasmid was used as a mock control. Two biological replicates were used per each group. Color represents gene expression from blue (downregulated) to red (upregulated). Fragment counts were normalized using the variance stabilizing transformation. (**B**) Generalized linear regression with counts modeled using negative binomial distribution was performed to generate MA plots (M—logarithmic fold change, A—mean of normalized counts). Genes are represented by dots; red color indicates significant change in expression between *IFNL4* vs. mock, *IFNL3* vs. mock and *IFNL4* vs. *IFNL3*. Dots on the upper part denote upregulated genes, whereas dots in the lower part indicate downregulated genes. Genes were considered differentially expressed when exhibiting an absolute log2-fold change of at least 1 and statistical significance. Note that IFNL4 and IFNL3 display identical gene expression profiles.

**Figure 5 ijms-22-02560-f005:**
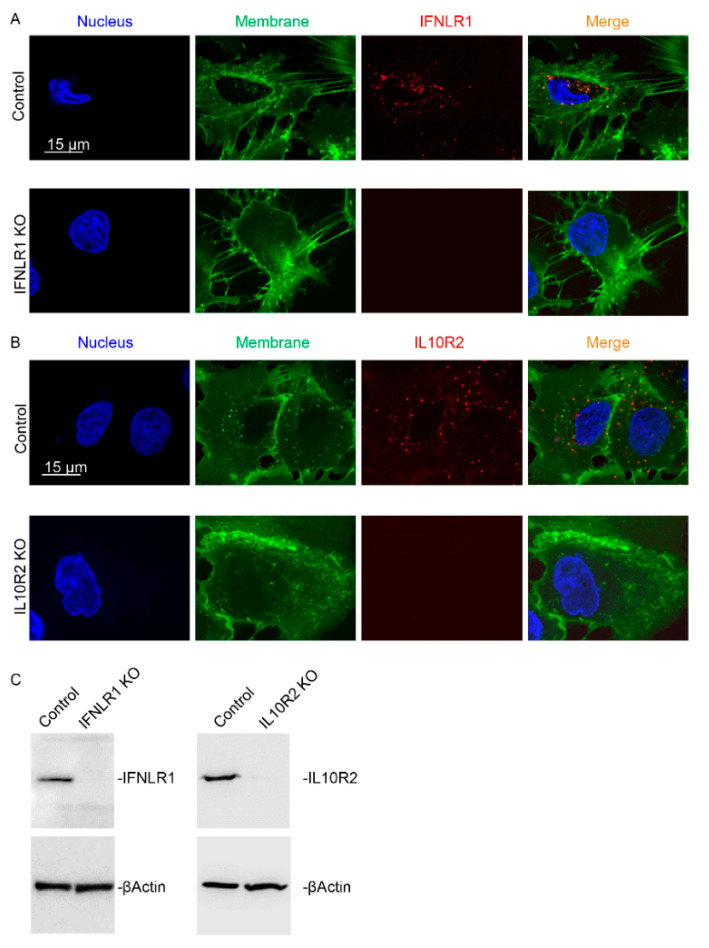
Characterization of IFNLR1 and IL10R2 knockout cell clones. High resolution confocal immunofluorescence images (**A**,**B**) and immunoblot analysis (**C**) of IFNLR1 and IL10R2 were performed in control and CRISPR/Cas9 edited HuH7 cells. Nuclei are shown in blue, membrane in green, IFNLR1 and IL10R2 are presented in red. Scale bar 15 µm. Representative pictures from several knockout clones are shown (**A**,**B**). An equal amount of several knockout clones was loaded per lane for Western blot analysis of IFNLR1 (*n* = 5) and IL10R2 (*n* = 2) (**C**).

**Figure 6 ijms-22-02560-f006:**
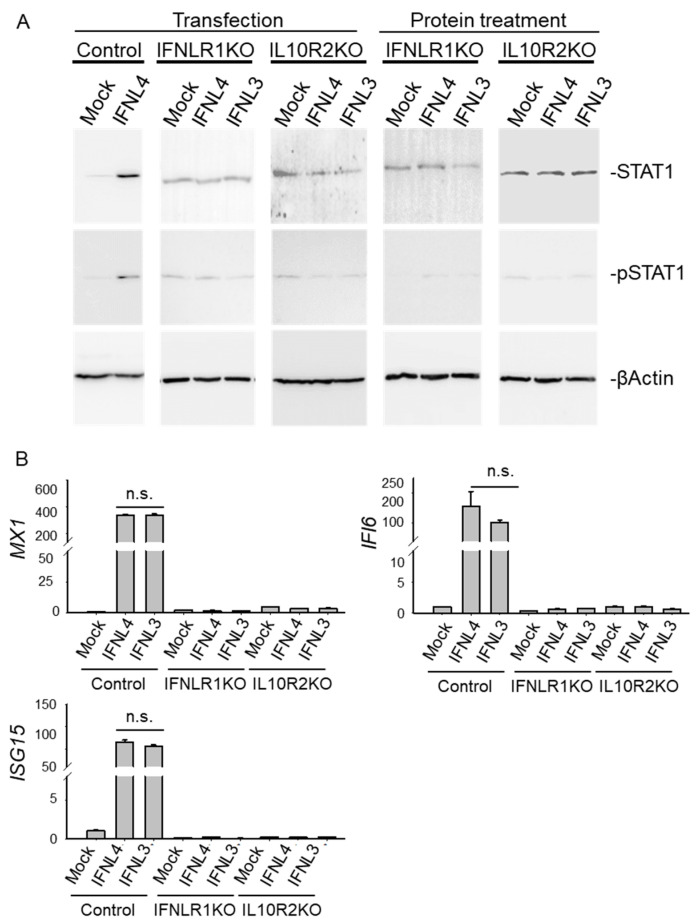
The IFNLR1/IL10R2 receptor complex is indispensable for the activation of JAK-STAT signaling triggered by IFNLs. (**A**) Wild type, IFNLR1 and IL10R2 knockout HuH7 cells were transfected with non-tagged *IFNL3* or *IFNL4* for 48 h or stimulated with recombinant IFNL3 or IFNL4 protein for 24 h. Empty plasmid was used as a mock control. Levels of STAT1 and pSTAT1 were determined by immunoblot analysis. Each lane is representative of 4 biological replicates. ßActin was used as a loading control. (**B**) Relative expression of *IFI6*, *MX1* and *ISG15* was determined by real time PCR in transfected cells. *GAPDH* was used as internal control. The data are presented as mean ± SEM (*n* ≥ 4).

**Figure 7 ijms-22-02560-f007:**
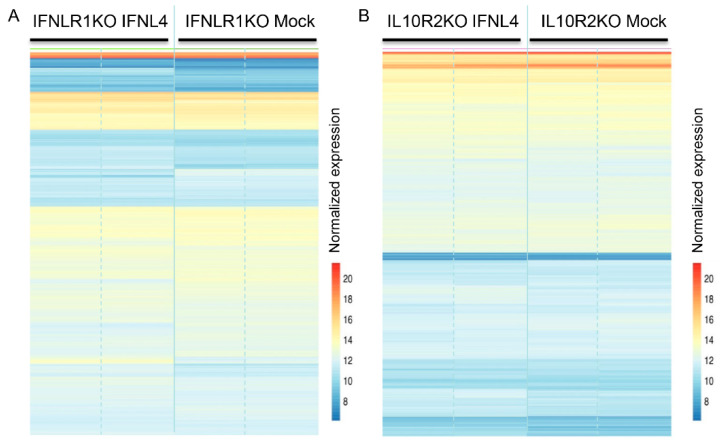
Gene expression profiles of IFNLR1 and IL10R2 KO cells transfected with *IFNL4.* Heat map of 1000 most expressed genes determined by RNAseq analysis of IFNLR1 −/− (**A**) and IL10R2 −/− (**B**) HuH7 cells transfected with non-tagged *IFNL4* for 48 h. Empty plasmid was used as a mock control. Two biological replicates were used per each group. Color represents gene expression from blue (downregulated) to red (upregulated). Fragment counts were normalized using the variance stabilizing transformation.

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
