# Peer review of "Expression of Interferons Lambda 3 and 4 Induces Identical Response in Human Liver Cell Lines Depending Exclusively on Canonical Signaling"

_ijms, 2021, doi:10.3390/ijms22052560_

Round 1

Reviewer 1 Report

The authors satisfactorily addressed my concerns.

Author Response

We would like to thank the Reviewer for his/her careful and rigorous review and constructive criticism that helped us to improve the quality of our paper.

Reviewer 2 Report

The Authors aimed to explore the anticipated non-canonical IFNL4-specific signaling in cells lacking either IFNLR1 or IL10R2 and a hypothetical intracellular pathway combining canonical JAK89 STAT signaling with post-receptor signal modification by IFNL4 expressed in cytosol. They stated that transfection with non-tagged interferons IFNL3 and IFNL4 equally upregulated STAT1 expression and phosphorylation whereas other previously published studies have shown that STAT1 expression and phosphorylation are higher in cells overexpressing tagged IFNL4. The Authors pointed that the hypothesis about IFNL4-specific non-canonical intra-, auto- or paracrine signaling very unlikely. The strength of the work is the research design, description of methods, statistical tests (ANOVA with Newman-Keuls test), discussion. The manuscript is a significant contribution to the scientific discussion about properties of interferons lambda.

Author Response

We would like to thank the Reviewer for his/her careful and rigorous review and constructive criticism that helped us to improve the quality of our paper.

To emphasize the novelty of our study, we modified the title, revised references and rephrased the last section of the introduction.

Reviewer 3 Report

Lunova et al examine whether IFNL3 and IFNL4 have redundant or distinct signaling pathways. To uncover this, they use transfection of either IFN into hepatic cell lines and then look for the upregulation of ISGs or the activation of pSTAT1 in the presence or absence of the IFN receptors. Using this strategy, they conclude that both IFNs are similar and IFNL4 does not have a unique intracellular pathway which is activated.

From the work presented I have several concerns and comments. From the introduction and discussion, it is not clear what is novel about this study. It seems that all work performed in the study has been already performed by other groups several years before. While it is good to confirm previous reports, it is also nice to extend them a bit which I found lacking in the current report. The use of two biological replicates for RNA-Seq data is also very concerning as it does not allow for proper statistics to be performed and a third replicate should be included to be able to make the correct statements here.  

Additionally, I found that the order of the manuscript made it difficult to follow. I would suggest that the author spend some time on restructuring the entire order to have a flow that makes more sense for the reader. Figure 1 is fine, Figure 2 A should become C. Then Figure 6 and 7 should be moved up to show that there is no cytosolic activity and that the activity only comes from the secreted protein, and that they produce similar signaling cascades. This would then be more logical why they want to remove the IFN receptors and test the effects of secreted IFNs. Figures 3-5 should be the last figures of the paper.

Major Points:

  1. The method of using transfected cells to see the response was also a bit strange. It was unclear why the supernatants were not harvested and used to look at effects. Additionally, the timings of 48h for transfection compared to 24h of treatment was not well controlled. There needs to be more controls added to show that these are relevant time points. 24h post-treatment is a very long time to look at pSTAT1 and needs to be more justified. The control was a qPCR, this does not show when and how long IFNs are secreted. Without showing an ELISA for IFN secretion or a functional assay for IFN secretion then there is no rational why 24h IFN protein treatment is a proper control.
  2. The statement in line 119 that the non-tagged IFNs induce the same STAT1 protein is confusing. The blots show that the protein treatment and the tagged IFNs show similar levels of STAT1 and pSTAT1.
  3. The order of Figure 2 is confusing. This should start with the time course and the ISGs and then finish with the pSTAT. As is, it does not make any sense why you pick 48h for looking at p-STAT and then you have to read the next section to understand. If they were inverted, it would provide a rationale for the experiment.
  4. I don't understand why the STAT1 levels are affected by removing IFNLR1 or IL10R2. These should only target the receptors. STAT should be unaffected as there are plenty of other receptors that it will still be functional an rely on STAT1 (i.e. IFNAR1/2). I understand that the activation of STAT will be changed but why are the basal levels of STAT1 so depleted? This seems like there is something wrong with the system. Additionally, in the HuH cells, there is no induction of pSTAT even in the wild type cells. It’s hard to conclude that there is an effect with the knock-out when there is no induction of the WT. The authors should check and see if they inverted the labeling of the blots in Figure 4 and explain why STAT1 is reduced.
  5. Figure 5: I don’t understand how a conclusion can be made about whether IFNL4 activates a unknown pathway when they are just compared to mock and not IFNL3. Also, with only two biological replicates used, how can any statistical information be gained? Figure 7 also used only two biological replicates which cannot be used to determine significance.

Minor points:

  1. Line 100 says that you test four ISGs but then you list eight ISGs. This makes is confusing to read. Please update to make it clearer that you compare eight in non-transfected and compare four of these with the transfected ones. Also, I don't understand why you show eight if you don’t compare them all. It would be easier to remove the ones from the supplemental figure and just show the four that are used to compare as the additional four do not bring any new information.
  2. Line 117 why are supplementary figures 2 and 3 cited? I think that only supplementary figure 2 should be cited here. It should also be cited in the text saying that quantification showed ….. otherwise it should be removed if it is not clear what it is bringing.
  3. Supplementary tables 1-3 were not included so I am unable to use them as a reference.
  4. Line 153, should read supplementary figure 4, correct?
  5. The order of supplementary figure 3 and 4 is wrong. You need to show the knock-out confirmation first and then show the results of how they respond to IFN. These should be inverted.
  6. Lines 176-178 are confusing and should be re-worded.
  7. Figure 6 is out of order. Should come earlier as a reason to justify the knock-out experiments.
  8. Line 253 should read Thus instead of This
  9. Line 264 should bead implies not imply
  10. Lines 276-278 contains a double negative and should be re-formatted
  11. Line 284-285 claims that RNA-Seq was performed in knock-out cells for both IFNl3 and IFNL4, however this not the data shown by the authors. In the paper RNA-Seq was only performed for IFNL4 with KO cells while they refer to Figure 7 which seems to show WT cells with IFNL3 and IFNL4. If Figure 7 does not have WT cells, it should be more clearly indicated in the text. This section should be re-written to properly reflect the paper.

Round 2

Reviewer 3 Report

The authors have made a minor effort to make any corrections to their paper. They could have made a slight effort to make things more clear but decided to just argue away things with text. While usually MDPI journals only give 10 days, they are very flexible and offer extensions when more time is needed to perform experiments. The authors could have done a few extra Western blots to validate their method.

Major concerns:

For figure 2. It was asked to change the order of the figure. The authors changed the order of the figure but kept the text the same. This makes no sense as the problem is the logic in the explanation. The part which is now 2A and B needs to be explained first to rationalize the second. I still find the timings to be strange and not well justified. The authors should have performed a few basic experiments to show that the timings make sense. I am very aware of how IFNs signal and their text argument is not sufficient. They need to show in their system that the timings they use make sense and that the timing of secretion of IFNL3 and L4 is equal. While they show that the qPCR looks similar, this does not mean that they are secreted with the same kinetics. I understand that ELISA can be hard for IFNL4 but there are alternative methods to use. They can simply repeat the same experiments that that have performed (Western blot for pSTAT1) but add several time points to show that the one they are using is relevant for both IFNL3 and IFNL4. Without this validation, the rest of their story is hard to say if it is correct. It is essential that this control is added to make sure that any conclusions can be made.  

The explanation that 2 samples is enough for RNA-Seq is based on a biology principle that they are homogenous. However, the issue is not whether the cells are homogeneous but about being able to make statically statements and statistics needs at least 3 values. As the authors make no statistic statements and their RNA-Seq shows nothing, two samples is enough to show nothing.

Section 2.3 about checking for cytosolic pathways needs more description as to why this is being tested. This comes out of nowhere now and needs to have a better introduction as to why this is controlled.

Fig 6 still is not convincing. It does not make sense that STAT1 is induced by IFNL3 and L4 where there is no effect on pSTAT1. Usually, STAT1 is used as a loading control to see pSTAT1 levels. If the point of this is to show that the receptors are really knock-ed out then you would need to see pSTAT-1 induction in the control cells from protein treatment. The current Western blot does not show this. Currently the KO and the wild type show the same lack of induction of pSTAT1. The authors need to show that protein treatment activates pSTAT1 (this is also lacking in figure 2) and makes the whole comparison difficult.

Round 3

Reviewer 3 Report

Lunova et al have improved their manuscript in this round. There are just a few minor points that are left open:

  1. The abstract and intro need to have some English editing. There are several places that are missing a “the” or “a”.
  2. Line 131 states that transfection induced similar STAT1 and pSTAT1 protein expression – this is not true. The quantification (Sup Fig 2) shows that transfected non-tagged proteins induced almost 50 times more pSTAT1 than protein treatment. If you are wanting to say that IFNL3 and IFNL4 are equal then you need to rephrase the sentence – something more like “IFNL3 and IFNL4 induced equal amounts of …” When you start the sentence with transfection and protein stimulation it seems that this is what you are comparing instead of IFNL3 vs IFNL4.
  3. The western blot provided in the answers to reviewers needs to be added to the text. This should be added either in Figure 2 or as a supplementary figure to show the timings of pSTAT1 secretion.

Author Response

We would like to thank the Reviewer for his/her careful and rigorous review and constructive criticism that helped us to improve the quality of our paper. Detailed point-by-point responses to all Reviewer’s remarks together with the corresponding amendments made to the manuscript are provided below. Reviewer’s questions and comments are given in italic; replies are given in blue and the changes in the manuscript are marked in red.

Reviewer #3 (Comments to the author):

Lunova et al have improved their manuscript in this round. There are just a few minor points that are left open:

1.The abstract and intro need to have some English editing. There are several places that are missing a “the” or “a”.

We thank the reviewer for paying attention to language editing. We found and added the missing terms „the“ or „a“ in the abstract / introduction section as recommended.

2.Line 131 states that transfection induced similar STAT1 and pSTAT1 protein expression – this is not true. The quantification (Sup Fig 2) shows that transfected non-tagged proteins induced almost 50 times more pSTAT1 than protein treatment. If you are wanting to say that IFNL3 and IFNL4 are equal then you need to rephrase the sentence – something more like “IFNL3 and IFNL4 induced equal amounts of …” When you start the sentence with transfection and protein stimulation it seems that this is what you are comparing instead of IFNL3 vs IFNL4.

We thank the reviewer for this comment. As suggested by the reviewer, we rephrased the text as follows:

“In the next step, we compared the ability of cells transfected with tagged and non-tagged IFNL3 and IFNL4 to activate JAK-STAT signaling by immunoblotting with densitometry quantification (Figure 2C and Supplementary Figures 2 and 4). HuH7 and HepG2 cells were transfected with tagged and non-tagged IFNL3 or IFNL4 for 48 hours or stimulated with IFNL3 or IFNL4 proteins for 24 hours. Transfection time of 48 hours was chosen based on our data (Supplementary Figure 1C) and on published data by Prokunina-Olsson et al. Transfection with non-tagged IFNL3 and non-tagged IFNL4 induced similar STAT1 protein expression and phosphorylation (pSTAT1); however, the response to transfection with Halo-tagged IFNL3 was significantly lower when compared to cells transfected with Halo-tagged IFNL4.

In a separate set of experiments conducted with cells stimulated by non-tagged recombinant interferons we observed that IFNL3-stimulated cells displayed similar levels of STAT1 and pSTAT1 to those detected in IFNL4-stimulated cells.”

3.The western blot provided in the answers to reviewers needs to be added to the text. This should be added either in Figure 2 or as a supplementary figure to show the timings of pSTAT1 secretion.

We agree with the reviewer and we included the figure to the revised draft as Supplementary Figure 1C.
